# Your classifier is secretly an energy based model and you should treat it like one

**Will Grathwohl**
University of Toronto & Vector Institute
Google Research
wgrathwohl@cs.toronto.edu

**Kuan-Chieh Wang**\***& Jörn-Henrik Jacobsen**\*
University of Toronto & Vector Institute
wangkua1@cs.toronto.edu
j.jacobsen@vectorinstitute.ai

**David Duvenaud**
University of Toronto & Vector Institute
duvenaud@cs.toronto.edu

**Kevin Swersky & Mohammad Norouzi**
Google Research
{kswersky, mnorouzi}@google.com

## ABSTRACT

We propose to reinterpret a standard discriminative classifier of $p(y|\mathbf{x})$ as an energy based model for the joint distribution $p(\mathbf{x}, y)$. In this setting, the standard class probabilities can be easily computed as well as unnormalized values of $p(\mathbf{x})$ and $p(\mathbf{x}|y)$. Within this framework, standard discriminative architectures may be used and the model can also be trained on unlabeled data. We demonstrate that energy based training of the joint distribution improves calibration, robustness, and out-of-distribution detection while also enabling our models to generate samples rivaling the quality of recent GAN approaches. We improve upon recently proposed techniques for scaling up the training of energy based models and present an approach which adds little overhead compared to standard classification training. Our approach is able to achieve performance rivaling the state-of-the-art in both generative and discriminative learning within one hybrid model.

## 1 INTRODUCTION

For decades, research on generative models has been motivated by the promise that generative models can benefit downstream problems such as semi-supervised learning, imputation of missing data, and calibration of uncertainty (*e.g.,* Chapelle et al. (2006); Dempster et al. (1977)). Yet, most recent research on deep generative models ignores these problems, and instead focuses on qualitative sample quality and log-likelihood on heldout validation sets.

Currently, there is a large performance gap between the strongest generative modeling approach to downstream tasks of interest and hand-tailored solutions for each specific problem. One potential explanation is that most downstream tasks are discriminative in nature and state-of-the-art generative models have diverged quite heavily from state-of-the-art discriminative architectures. Thus, even when trained solely as classifiers, the performance of generative models is far below the performance of the best discriminative models. Hence, the potential benefit from the generative component of the model is far outweighed by the decrease in discriminative performance. Recent work (Behrmann et al., 2018; Chen et al., 2019) attempts to improve the discriminative performance of generative models by leveraging invertible architectures, but these methods still underperform their purely discriminative counterparts jointly trained as generative models.

This paper advocates the use of energy based models (EBMs) to help realize the potential of generative models on downstream discriminative problems. While EBMs are currently challenging to work with, they fit more naturally within a discriminative framework than other generative models and facilitate the use of modern classifier architectures. Figure 1 illustrates an overview of the architecture, where the logits of a classifier are re-interpreted to define the joint density of data points and labels and the density of data points alone.

---

\*Equal Contribtuion

The contributions of this paper can be summarized as: 1) We present a novel and intuitive framework for joint modeling of labels and data. 2) Our models considerably outperform previous state-of-the-art hybrid models at both generative and discriminative modeling. 3) We show that the incorporation of generative modeling gives our models improved calibration, out-of-distribution detection, and adversarial robustness, performing on par with or better than hand-tailored methods for multiple tasks.

## 2 ENERGY BASED MODELS

Energy based models (LeCun et al., 2006) hinge on the observation that any probability density $p(\mathbf{x})$ for $\mathbf{x} \in \mathbb{R}^D$ can be expressed as

$$p_\theta(\mathbf{x}) = \frac{\exp(-E_\theta(\mathbf{x}))}{Z(\theta)} , \qquad (1)$$

where $E_\theta(\mathbf{x}) : \mathbb{R}^D \to \mathbb{R}$, known as the *energy function*, maps each point to a scalar, and $Z(\theta) = \int_{\mathbf{x}} \exp(-E_\theta(\mathbf{x}))$ is the normalizing constant (with respect to $\mathbf{x}$) also known as the partition function. Thus, one can parameterize an EBM using any function that takes $\mathbf{x}$ as the input and returns a scalar.

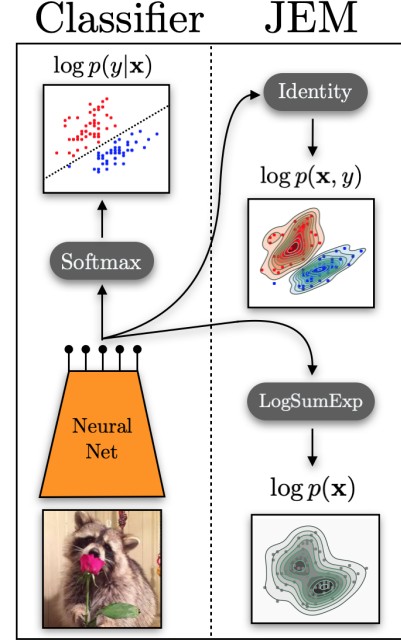

Figure 1: Visualization of our method, JEM, which defines a joint EBM from classifier architectures.

For most choices of $E_\theta$, one cannot compute or even reliably estimate $Z(\theta)$, which means estimating normalized densities is intractable and standard maximum likelihood estimation of the parameters, $\theta$, is not straightforward. Thus, we must rely on other methods to train EBMs. We note that the derivative of the log-likelihood for a single example $\mathbf{x}$ with respect to $\theta$ can be expressed as

$$\frac{\partial \log p_\theta(\mathbf{x})}{\partial \theta} = \mathbb{E}_{p_\theta(\mathbf{x}')}\left[ \frac{\partial E_\theta(\mathbf{x}')}{\partial \theta} \right] - \frac{\partial E_\theta(\mathbf{x})}{\partial \theta} , \qquad (2)$$

where the expectation is over the model distribution. Unfortunately, we cannot easily draw samples from $p_\theta(\mathbf{x})$, so we must resort to MCMC to use this gradient estimator. This approach was used to train some of the earliest EBMs. For example, Restricted Boltzmann Machines (Hinton, 2002) were trained using a block Gibbs sampler to approximate the expectation in Eq. (2).

Despite a long period of little development, there has been recent work using this method to train large-scale EBMs on high-dimensional data, parameterized by deep neural networks (Nijkamp et al., 2019b;a; Du & Mordatch, 2019; Xie et al., 2016). These recent successes have approximated the expectation in Eq. (2) using a sampler based on Stochastic Gradient Langevin Dynamics (SGLD) (Welling & Teh, 2011) which draws samples following

$$\mathbf{x}_0 \sim p_0(\mathbf{x}), \qquad \mathbf{x}_{i+1} = \mathbf{x}_i - \frac{\alpha}{2} \frac{\partial E_\theta(\mathbf{x}_i)}{\partial \mathbf{x}_i} + \epsilon, \qquad \epsilon \sim \mathcal{N}(0, \alpha) \qquad (3)$$

where $p_0(\mathbf{x})$ is typically a Uniform distribution over the input domain and the step-size $\alpha$ should be decayed following a polynomial schedule. In practice the step-size, $\alpha$, and the standard deviation of $\epsilon$ is often chosen separately leading to a biased sampler which allows for faster training. See Appendix H.1 for further discussion of samplers for EBM training.

## 3 WHAT YOUR CLASSIFIER IS HIDING

In modern machine learning, a classification problem with $K$ classes is typically addressed using a parametric function, $f_\theta : \mathbb{R}^D \to \mathbb{R}^K$, which maps each data point $\mathbf{x} \in \mathbb{R}^D$ to $K$ real-valued numbers known as logits. These logits are used to parameterize a categorical distribution using the so-called Softmax transfer function:

$$p_\theta(y \mid \mathbf{x}) = \frac{\exp\left(f_\theta(\mathbf{x})[y]\right)}{\sum_{y'} \exp\left(f_\theta(\mathbf{x})[y']\right)} , \qquad (4)$$

where $f_\theta(\mathbf{x})[y]$ indicates the $y^{\text{th}}$ index of $f_\theta(\mathbf{x})$, *i.e.,* the logit corresponding the the $y^{\text{th}}$ class label.

Our key observation in this work is that one can slightly re-interpret the logits obtained from $f_\theta$ to define $p(\mathbf{x}, y)$ and $p(\mathbf{x})$ as well. Without changing $f_\theta$, one can re-use the logits to define an energy based model of the joint distribution of data point $\mathbf{x}$ and labels $y$ via:

$$p_\theta(\mathbf{x}, y) = \frac{\exp\left(f_\theta(\mathbf{x})[y]\right)}{Z(\theta)} , \tag{5}$$

where $Z(\theta)$ is the unknown normalizing constant and $E_\theta(\mathbf{x}, y) = -f_\theta(\mathbf{x})[y]$.

By marginalizing out $y$, we obtain an unnormalized density model for $\mathbf{x}$ as well,

$$p_\theta(\mathbf{x}) = \sum_y p_\theta(\mathbf{x}, y) = \frac{\sum_y \exp\left(f_\theta(\mathbf{x})[y]\right)}{Z(\theta)} . \tag{6}$$

Notice now that the LogSumExp$(\cdot)$ of the logits of *any* classifier can be re-used to define the energy function at a data point $\mathbf{x}$ as

$$E_\theta(\mathbf{x}) = -\text{LogSumExp}_y(f_\theta(\mathbf{x})[y]) = -\log \sum_y \exp(f_\theta(\mathbf{x})[y]) . \tag{7}$$

Unlike typical classifiers, where shifting the logits $f_\theta(\mathbf{x})$ by an arbitrary scalar does not affect the model at all, in our framework, shifting the logits for a data point $\mathbf{x}$ will affect $\log p_\theta(\mathbf{x})$. Thus, we are making use of the extra degree of freedom hidden within the logits to define the density function over input examples as well as the joint density among examples and labels. Finally, when we compute $p_\theta(y \mid \mathbf{x})$ via $p_\theta(\mathbf{x}, y)/p_\theta(\mathbf{x})$ by dividing Eq. (5) to Eq. (6), the normalizing constant cancels out, yielding the standard Softmax parameterization in Eq. (4). Thus, we have found a generative model hidden within every standard discriminative model! Since our approach proposes to reinterpret a classifier as a **J**oint **E**nergy-based **M**odel we refer to it throughout this work as JEM.

## 4 OPTIMIZATION

We now wish to take advantage of our new interpretation of classifier architectures to gain the benefits of generative models while retaining strong discriminative performance. Since our model's parameterization of $p(y|\mathbf{x})$ is normalized over $y$, it is simple to maximize its likelihood as in standard classifier training. Since our models for $p(\mathbf{x})$ and $p(\mathbf{x}, y)$ are unnormalized, maximizing their likelihood is not as easy. There are many ways we could train $f_\theta$ to maximize the likelihood of the data under this model. We could apply the gradient estimator of Equation 2 to the likelihood under the joint distribution of Equation 5. Using Equations 6 and 4, we can also factor the likelihood as

$$\log p_\theta(\mathbf{x}, y) = \log p_\theta(\mathbf{x}) + \log p_\theta(y|\mathbf{x}). \tag{8}$$

The estimator of Equation 2 is biased when using a MCMC sampler with a finite number of steps. Given that the goal of our work is to incorporate EBM training into the standard classification setting, the distribution of interest is $p(y|\mathbf{x})$. For this reason we propose to train using the factorization of Equation 8 to ensure this distribution is being optimized with an unbiased objective. We optimize $p(y|\mathbf{x})$ using standard cross-entropy and optimize $\log p(\mathbf{x})$ using Equation 2 with SGLD where gradients are taken with respect to $\text{LogSumExp}_y(f_\theta(x)[y])$. We find alternative factorings of the likelihood lead to considerably worse performance as can be seen in Section 5.1.

Following Du & Mordatch (2019) we use persistent contrastive divergence (Tieleman, 2008) to estimate the expectation in the right-hand-side of Equation 2 since it gives an order of magnitude savings in computation compared to seeding new chains at each iteration as in Nijkamp et al. (2019b). This comes at the cost of decreased training stability. These trade-offs are discussed in Appendix H.2.

## 5 APPLICATIONS

We completed a thorough empirical investigation to demonstrate the benefits of JEM over standard classifiers. First, we achieved performance rivaling the state of the art in *both* discriminative and generative modeling. Even more interesting, we observed a number of benefits related to the practical application of discriminative models including improved uncertainty quantification, out-of-distribution detection, and robustness to adversarial examples. Generative models have been long-expected to provide these benefits but have never been demonstrated to do so at this scale.

| Class | Model | Accuracy% ↑ | IS↑ | FID↓ |
|-------|-------|-------------|-----|------|
| **Hybrid** | Residual Flow | 70.3 | 3.6 | 46.4 |
| | Glow | 67.6 | 3.92 | 48.9 |
| | IGEBM | 49.1 | 8.3 | **37.9** |
| | JEM $p(\mathbf{x}|y)$ factored | 30.1 | 6.36 | 61.8 |
| | JEM (Ours) | **92.9** | **8.76** | 38.4 |
| **Disc.** | Wide-Resnet | 95.8 | N/A | N/A |
| **Gen.** | SNGAN | N/A | 8.59 | 25.5 |
| | NCSN | N/A | 8.91 | 25.32 |

Table 1: CIFAR10 Hybrid modeling Results. Residual Flow (Chen et al., 2019), Glow (Kingma & Dhariwal, 2018), IGEBM (Du & Mordatch, 2019), SNGAN (Miyato et al., 2018), NCSN (Song & Ermon, 2019)

Figure 2: CIFAR10 class-conditional samples.

All architectures used are based on Wide Residual Networks (Zagoruyko & Komodakis, 2016) where we have removed batch-normalization[1] to ensure that our models' outputs are deterministic functions of the input. This slightly increases classification error of a WRN-28-10 from $4.2\%$ to $6.4\%$ on CIFAR10 and from $2.3$ to $3.4\%$ on SVHN.

All models were trained in the same way with the same hyper-parameters which were tuned on CIFAR10. Intriguingly, the SGLD sampler parameters found here generalized well across datasets and model architectures. All models are trained on a single GPU in approximately 36 hours. Full experimental details can be found in Appendix A.

## 5.1 HYBRID MODELING

First, we show that a given classifier architecture can be trained as an EBM to achieve competitive performance as both a classifier and a generative model. We train JEM on CIFAR10, SVHN, and CIFAR100 and compare against other hybrid models as well as standalone generative and discriminative models. We find JEM performs near the state of the art in both tasks simultaneously, outperforming other hybrid models (Table 1).

Given that we cannot compute normalized likelihoods, we present inception scores (IS) (Salimans et al., 2016) and Frechet Inception Distance (FID) (Heusel et al., 2017) as a proxy for this quantity. We find that JEM is competitive with SOTA generative models at these metrics. These metrics are not commonly reported on CIFAR100 and SVHN so we present accuracy and qualitative samples on these datasets. Our models achieve $96.7\%$ and $72.2\%$ accuracy on SVHN and CIFAR100, respectively. Samples from JEM can be seen in Figures 2, 3 and in Appendix C.

SVHN

CIFAR100

Figure 3: Class-conditional samples.

JEM is trained to maximize the likelihood factorization shown in Eq. 8. This was to ensure that no bias is added into our estimate of $\log p(y|\mathbf{x})$ which can be computed exactly in our setup. Prior work (Du & Mordatch, 2019; Xie et al., 2016) proposes to factorize the objective as $\log p(\mathbf{x}|y) + \log p(y)$. In these works, each $p(\mathbf{x}|y)$ is a separate EBM with a distinct, unknown normalizing constant, meaning that their model cannot be used to compute $p(y|\mathbf{x})$ or $p(\mathbf{x})$. This explains why the model of Du & Mordatch (2019) (we will refer to this model as IGEBM) is not a competitive classifier. As an ablation, we trained JEM to maximize this objective and found a considerable decrease in discriminative performance (see Table 1, row 4).

## 5.2 CALIBRATION

A classifier is considered calibrated if its predictive confidence, $\max_y p(y|\mathbf{x})$, aligns with its misclassification rate. Thus, when a calibrated classifier predicts label $y$ with confidence .9 it should have a $90\%$ chance of being correct. This is an important feature for a model to have when deployed in real-world scenarios where outputting an incorrect decision can have catastrophic consequences. The classifier's confidence can be used to decide when to output a prediction or deffer to a human,

---

[1]This was done to remove sources of stochasticity in early experiments. Since then we have been able to successfully train Joint-EBMs with Batch Normalization and other forms of stochastic regularization (such as dropout) without issue. We leave the incorporation of these methods to further work.

for example. Here, a well-calibrated, but less accurate classifier can be considerably more useful than a more accurate, but less-calibrated model.

While classifiers have grown more accurate in recent years, they have also grown considerably less calibrated (Guo et al., 2017). Contrary to this behavior, we find that JEM notably improves classification while retaining high accuracy.

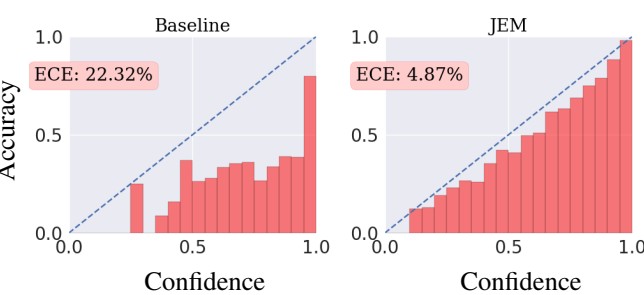

We focus on CIFAR100 since SOTA classifiers achieve approximately 80% accuracy. We train JEM on this dataset and compare to a baseline of the same architecure without EBM training. Our baseline model achieves

Figure 4: CIFAR100 calbration results. ECE = Expected Calibration Error (Guo et al., 2017), see Appendix E.1.

74.2% accuracy and JEM achieves 72.2% (for reference, a ResNet-110 achieves 74.8% accuracy (Zagoruyko & Komodakis, 2016)). We find the baseline model is very poorly calibrated outputting highly over-confident predictions. Conversely, we find JEM produces a nearly perfectly calibrated classifier when measured with Expected Calibration Error (see Appendix E.1). Compared to other calibration methods such as Platt scaling (Guo et al., 2017), JEM requires no additional training data. Results can be seen in Figure 4 and additional results can be found in Appendix E.2.

## 5.3 OUT-OF-DISTRIBUTION DETECTION

In general, out-of-distribution (OOD) detection is a binary classification problem, where the model is required to produce a score

$$s_\theta(\mathbf{x}) \in \mathbb{R},$$

where $\mathbf{x}$ is the query, and $\theta$ is the set of learnable parameters. We desire that the scores for in-distribution examples are higher than that out-of-distribution examples. Typically for evaluation, threshold-free metrics are used, such as the area under the receiver-operating curve (AUROC) (Hendrycks & Gimpel, 2016). There exist a number of distinct OOD detection approaches to which JEM can be applied. We expand on them below. Further results and experimental details can be found in Appendix F.2.

### 5.3.1 INPUT DENSITY

A natural approach to OOD detection is to fit a density model on the data and consider examples with low likelihood to be OOD. While intuitive, this approach is currently not competitive on high-dimensional data. Nalisnick et al. (2018) showed that tractable deep generative models such as Kingma & Dhariwal (2018) and Salimans et al. (2017) can assign higher densities to OOD examples than in-distribution examples. Further work (Nalisnick et al., 2019) shows examples where the densities of an OOD dataset are completely indistinguishable from the in-distribution set, *e.g.,* see Table 2, column 1. Conversely, Du & Mordatch (2019) have shown that the likelihoods from EBMs can be reliably used as a predictor for OOD inputs. As can be seen in Table 2 column 2, JEM consistently assigns higher likelihoods to in-distribution data than OOD data. One possible explanation for JEM's further improvement over IGEBM is its ability to incorporate labeled information during training while also being able to derive a principled model of $p(\mathbf{x})$. Intriguingly, Glow does not appear to benefit in the same way from this supervision as is demonstrated by the little difference between our unconditional and class-conditional Glow results. Quantitative results can be found in Table 3 (top).

### 5.3.2 PREDICTIVE DISTRIBUTION

Many successful approaches have utilized a classifier's predictive distribution for OOD detection (Gal & Ghahramani, 2016; Wang et al., 2018; Liang et al., 2017). A useful OOD score that can be derived from this distribution is the maximum prediction probability: $s_\theta(\mathbf{x}) = \max_y p_\theta(y|\mathbf{x})$

(Hendrycks & Gimpel, 2016). It has been demonstrated that OOD performance using this score is highly correlated with a model's classification accuracy. Since JEM is a competitive classifier, we find it performs on par (or beyond) the performance of a strong baseline classifier and considerably outperforms other generative models. Results can be seen in Table 3 (middle).

### 5.3.3 A NEW SCORE: APPROXIMATE MASS

It has been recently proposed that likelihood may not be enough for OOD detection in high dimensions (Nalisnick et al., 2019). It is possible for a point to have high likelihood under a distribution yet be nearly impossible to be sampled. Real samples from a distribution lie in what is known as the "typical" set. This is the area of high probability *mass*. A single point may have high density but if the surrounding areas have very low density, then that point is likely not in the typical set and therefore likely not a sample from the data distribution. For a high-likelihood datapoint outside of the typical set, we expect the density to change rapidly around it, thus the norm of the gradient of the log-density will be large compared to examples in the typical set (otherwise it would be in an area of high mass). We propose an alternative OOD score based on this quantity:

$$s_\theta(\mathbf{x}) = - \left\| \frac{\partial \log p_\theta(\mathbf{x})}{\partial \mathbf{x}} \right\|_2 . \tag{9}$$

For EBMs (JEM and IGEBM), we find this predictor greatly outperforms our own and other generative model's likelihoods – see Table 2 column 3. For tractable likelihood methods we find this predictor is anti-correlated with the model's likelihood and neither is reliable for OOD detection. Results can be seen in Table 3 (bottom).

### 5.4 ROBUSTNESS

Recent work (Athalye et al., 2017) has demonstrated that classifiers trained to be adversarially robust can be re-purposed to generate convincing images, do in-painting, and translate examples from one class to another. This is done through an iterative refinement procedure, quite similar to the SGLD used to sample from EBMs. We also note that adversarial training (Goodfellow et al., 2014) bears many similarities to SGLD training of EBMs. In both settings, we use a gradient-based optimization procedure to generate examples which activate a specific high-level network activation, then optimize the weights of the network to minimize the generated example's effect on that activation. Further connections have been drawn between adversarial training and regularizing the gradients of the network's activations around the data (Simon-Gabriel et al., 2018). This is similar to the objective of Score Matching (Hyvärinen, 2005) which can also be used to train EBMs (Kingma & Lecun, 2010; Song & Ermon, 2019).

Given these connections one may wonder if a classifier derived from an EBM would be more robust to adversarial examples than a standard model. This behavior has been demonstrated in prior work on EBMs (Du & Mordatch, 2019) but their work did not produce a competitive discriminative model

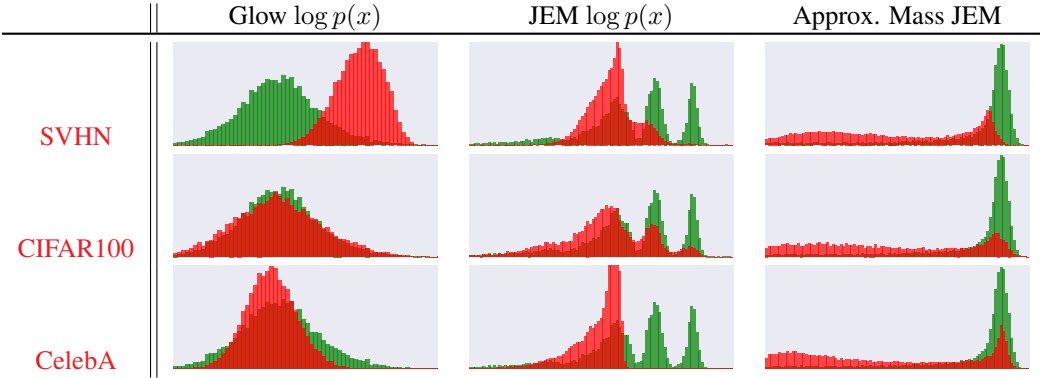

Table 2: Histograms for OOD detection. All models trained on CIFAR10. Green corresponds to the score on (in-distribution) CIFAR10, and red corresponds to the score on the OOD dataset.

| $s_\theta(\mathbf{x})$ | Model | CIFAR10 SVHN | CIFAR10 Interp | CIFAR100 | CelebA |
|---|---|---|---|---|---|
| $\log p(\mathbf{x})$ | Unconditional Glow | .05 | .51 | .55 | .57 |
| | Class-Conditional Glow | .07 | .45 | .51 | .53 |
| | IGEBM | .63 | **.70** | .50 | .70 |
| | JEM (Ours) | **.67** | .65 | **.67** | **.75** |
| $\max_y p(y\|\mathbf{x})$ | Wide-ResNet | **.93** | **.77** | .85 | .62 |
| | Class-Conditional Glow | .64 | .61 | .65 | .54 |
| | IGEBM | .43 | .69 | .54 | .69 |
| | JEM (Ours) | .89 | .75 | **.87** | **.79** |
| $\left\|\left\|\frac{\partial \log p(\mathbf{x})}{\partial \mathbf{x}}\right\|\right\|$ | Unconditional Glow | **.95** | .27 | .46 | .29 |
| | Class-Conditional Glow | .47 | .01 | .52 | .59 |
| | IGEBM | .84 | .65 | .55 | .66 |
| | JEM (Ours) | .83 | **.78** | **.82** | **.79** |

Table 3: OOD Detection Results. Models trained on CIFAR10. Values are AUROC.

and is therefore of limited practical application for this purpose. Similarly, we find JEM achieves considerable robustness without sacrificing discriminative performance.

### 5.4.1 IMPROVED ROBUSTNESS THROUGH EBM TRAINING

A common threat model for adversarial robustness is that of perturbation-based adversarial examples with an $L_p$-norm constraint (Goodfellow et al., 2014). They are defined as perturbed inputs $\tilde{\mathbf{x}} = \mathbf{x} + \delta$, which change a model's prediction subject to $||\tilde{\mathbf{x}} - \mathbf{x}||_p < \epsilon$. These examples exploit semantically meaningless perturbations to which the model is overly sensitive. However, closeness to real inputs in terms of a given metric does not imply that adversarial examples reside within areas of high density according to the model distribution, hence it is not surprising that the model makes mistakes when asked to classify inputs it has rarely or never encountered during training.

This insight has been used to detect and robustly classify adversarial examples with generative models (Song et al., 2017; Li et al., 2018; Fetaya et al., 2019). The state-of-the-art method for adversarial robustness on MNIST classifies by comparing an input to samples generated from a class-conditional generative model (Schott et al., 2018). This can be thought of as classifying an example similar to the input but from an area of higher density under the model's learned distribution. This refined input resides in areas where the model has already "seen" sufficient data and is thus able to accurately classify. Albeit promising, this family of methods has not been able to scale beyond MNIST due to a lack of sufficiently powerful conditional generative models. We believe JEM can help close this gap. We propose to run a few iterations of our model's sampling procedure seeded at

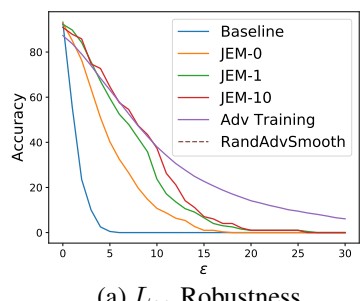

(a) $L_\infty$ Robustness

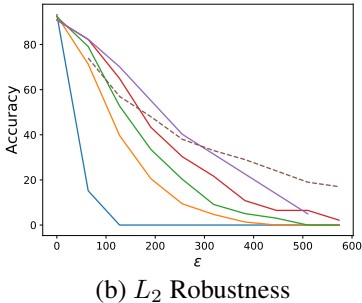

(b) $L_2$ Robustness

Figure 5: Adversarial Robustness Results with PGD attacks. JEM adds considerable robustness.

a given input. This should be able to transform low-probability inputs to a nearby point of high probability, "undoing" any adversarial attack and enabling the model to classify robustly.

**Perturbation Robustness** We run a number of powerful adversarial attacks on our CIFAR10 models. We run a white-box PGD attack, giving the attacker access to the gradients through our sampling procedure[2]. Because our sampling procedure is stochastic, we compute the "expectation over transformations" Athalye et al. (2018), the expected gradient over multiple runs of the sampling

---

[2] In Du & Mordatch (2019) the attacker was not given access to the gradients of the refinement procedure. We re-run these stronger attacks on their model as well and provide a comparison in Appendix G.

procedure. We also run gradient-free black-box attacks; the boundary attack (Brendel et al., 2017) and the brute-force pointwise attack (Rauber et al., 2017). All attacks are run with respect to the $L_2$ and $L_\infty$ norms and we test JEM with 0, 1, and 10 steps of sampling seeded at the input.

Results from the PGD experiments can be seen in Figure 5. Experimental details and remaining results, including gradient-free attacks, can be found in Appendix G. Our model is considerably more robust than a baseline with standard classifier training. With respect to both norms, JEM delivers considerably improved robustness when compared to the baseline but for many epsilons falls below state-of-the-art adversarial training (Madry et al., 2017; Santurkar et al., 2019) and the state-of-the-art certified robutness method of Salman et al. (2019) ("RandAdvSmooth" in Figure 5). We note that each of these baseline methods is trained to be robust to the norm through which it is being attacked and it has been shown that attacking an $L_\infty$ adversarially trained model with an $L_2$ adversary decreases robustness considerably (Madry et al., 2017). However, we attack the same JEM model with both norms and observe competitive robustness in both cases.

JEM with 0 steps refinement is noticeably more robust than the baseline model trained as a standard classifier, thus simply adding EBM training can produce more robust models. We also find that increasing the number of refinement steps further increases robustness to levels at robustness-specific approaches. We expect that increasing the number of refinement steps will lead to more robust models but due to computational constraints we could not run attacks in this setting.

**Distal Adversarials**   Another common failure mode of non-robust models is their tendency to classify non-sensical inputs with high confidence. To analyze this property, we follow Schott et al. (2018). Starting from noise we generate images to maximize $p(y = \text{"car"}|\mathbf{x})$. Results are shown in figure 6. The baseline confidently classifies unstructured noise images. The $L_2$ adversarially trained ResNet with $\epsilon = 0.5$ (Santurkar et al., 2019) confidently classifies somewhat structured, but unrealistic images. JEM does not confidently classify nonsensical images, so instead, car attributes and natural image properties visibly emerge.

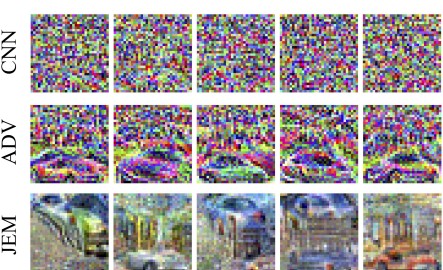

Figure 6: **Distal Adversarials.**   Confidently classified images generated from noise, such that: $p(y = \text{"car"}|\mathbf{x}) > .9$.

## 6   LIMITATIONS

Energy based models can be very challenging to work with. Since normalized likelihoods cannot be computed, it can be hard to verify that learning is taking place at all. When working in domains such as images, samples can be drawn and checked to assess learning, but this is far from a generalizable strategy. Even so, these samples are only samples from an approximation to the model so they can only be so useful. Furthermore, the gradient estimators we use to train JEM are quite unstable and are prone to diverging if the sampling and optimization parameters are not tuned correctly. Regularizers may be added (Du & Mordatch, 2019) to increase stability but it is not clear what effect they have on the final model. The models used to generate the results in this work regularly diverged throughout training, requiring them to be restarted with lower learning rates or with increased regularization. See Appendix H.3 for a detailed description of how these difficulties were handled.

While this may seem prohibitive, we believe the results presented in this work are sufficient to motivate the community to find solutions to these issues as any improvement in the training of energy based models will further improve the results we have presented in this work.

## 7   RELATED WORK

Prior work (Xie et al., 2016) made a similar observation to ours about classifiers and EBMs but define the model differently. They reinterpret the logits to define a class-conditional EBM $p(\mathbf{x}|y)$, similar to Du & Mordatch (2019). This setting requires additional parameters to be learned to derive a classifier and an unconditional model. We believe this subtle distinction is responsible for our model's success. The model of (Song & Ou, 2018) is similar as well but is trained using a GAN-like generator and is applied to different applications. Also related are Introspective Networks (Jin et al., 2017; Lee et al., 2018) which have drawn a similar connection between discriminative classifiers and generative models. They derive a generative model from a classifier which learns to distinguish

between data and negative examples generative via an MCMC-like procedure. Training in this way has also been shown to improve adversarial robustness.

Our work builds heavily on Nijkamp et al. (2019b;a); Du & Mordatch (2019) which scales the training of EBMs to high-dimensional data using Contrastive Divergence and SGLD. While these works have pushed the boundaries of the types of data to which we can apply EBMs, many issues still exist. These methods require many steps of SGLD to take place at each training iteration. Each step requires approximately the same amount of computation as one iteration of standard discriminitive model training, therefore training EBMs at this scale is orders of magnitude slower than training a classifier – limiting the size of problems we can attack with these methods. There exist orthogonal approaches to training EBMs which we believe have promise to scale more gracefully.

Score matching (Hyvärinen, 2005) attempts to match the derivative of the model's density with the derivative of the data density. This approach saw some development towards high-dimensional data (Kingma & Lecun, 2010) and recently has been successfully applied to large natural images (Song & Ermon, 2019). This approach required a model to output the derivatives of the density function, not the density function itself, so it is unclear what utility this model can provide to the applications we have discussed in this work. Regardless, we believe this is a promising avenue for further research. Noise Contrastive Estimation (Gutmann & Hyvärinen, 2010) rephrases the density estimation problem as a classification problem, attempting to distinguish data from a known noise distribution. If the classifier is properly structured, then once the classification problem is solved, an unnormalized density estimator can be derived from the classifier and noise distribution. While this method has been recently extended (Ceylan & Gutmann, 2018), these methods are challenging to extend to high-dimensional data.

## 8 Conclusion and Further Work

In this work we have presented JEM, a novel reinterpretation of standard classifier architectures which retains the strong performance of SOTA discriminative models while adding the benefits of generative modeling approaches. Our work is enabled by recent work scaling techniques for training EBMs to high dimensional data. We have demonstrated the utility of incorporating this type of training into discriminative models. While there exist many issues in training EBMs we hope the results presented here will encourage the community to improve upon current approaches.

## 9 Acknowledgements

We would like to thank Ying Nian Wu and Mitch Hill for providing some EBM training tips and tricks which were crucial in getting this project off the ground. We would also like to thank Jeremy Cohen for his useful feedback which greatly strengthened our adversarial robustness results. We would like to thank Lukas Schott for feedback on the robustness evaluation, Alexander Meinke and Francesco Croce for spotting some typos and suggesting the transfer attack. We would also like to thank Zhuowen Tu and Kwonjoon Lee for bringing related work to our attention.

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

## A   TRAINING DETAILS

We train all models with the Adam optimizer (Kingma & Ba, 2014) for 150 epochs through the dataset using a staircase decay schedule. All network architecutres are based on WideResNet-28-10 with no batch normalization. We generate samples using PCD with hyperparameters in Table 4. We evolve the chains with 20-steps of SGLD per iteration and with probability .05 we reiniatilize the chains with uniform random noise. For preprocessing, we scale images to the range $[-1, 1]$ and add Gaussian noise of stddev = .03. Pseudo-code for our training procedure is in Algorithm 1.

When training via contrastive divergence there are a few different ways one could potentially draw samples from $p_\theta(\mathbf{x})$. We could:

1. Sample $y \sim p(y)$ then sample $\mathbf{x} \sim p_\theta(\mathbf{x}|y)$ via SGLD with energy $E(\mathbf{x}|y) = -f_\theta(\mathbf{x})[y]$ then throw away $y$.

2. Sample $\mathbf{x} \sim p_\theta(\mathbf{x})$ via SGLD with energy $E(x) = -\text{LogSumExp}_y f_\theta(\mathbf{x})[y]$.

We experimented with both methods during training and found that while method 1 produced more visually appealing samples (from a human's perspective), method 2 produced slightly stronger discirminative performance – 92.9% vs. 91.2% accuracy on CIFAR10. For this reason we use method 2 in all results presented.

---

**Algorithm 1** JEM training: Given network $f_\theta$, SGLD step-size $\alpha$, SGLD noise $\sigma$, replay buffer $B$, SGLD steps $\eta$, reinitialization frequency $\rho$

1: **while** not converged **do**
2:     Sample $\mathbf{x}$ and $y$ from dataset
3:     $L_{\text{clf}}(\theta) = \text{xent}(f_\theta(\mathbf{x}), y)$
4:     Sample $\widehat{\mathbf{x}}_0 \sim B$ with probability $1 - \rho$, else $\widehat{\mathbf{x}}_0 \sim \mathcal{U}(-1, 1)$         ▷ Initialize SGLD
5:     **for** $t \in [1, 2, \ldots, \eta]$ **do**         ▷ SGLD
6:         $\widehat{\mathbf{x}}_t = \widehat{\mathbf{x}}_{t-1} + \alpha \cdot \frac{\partial \text{LogSumExp}_{y'}(f_\theta(\widehat{\mathbf{x}}_{t-1})[y'])}{\partial \widehat{\mathbf{x}}_{t-1}} + \sigma \cdot \mathcal{N}(0, I)$
7:     **end for**
8:     $L_{\text{gen}}(\theta) = \text{LogSumExp}_{y'}(f(\mathbf{x})[y']) - \text{LogSumExp}_{y'}(f(\widehat{\mathbf{x}}_t)[y'])$         ▷ Surrogate for Eq 2
9:     $L(\theta) = L_{\text{clf}}(\theta) + L_{\text{gen}}(\theta)$
10:     Obtain gradients $\frac{\partial L(\theta)}{\partial \theta}$ for training
11:     Add $\widehat{\mathbf{x}}_t$ to $B$
12: **end while**

---

| Variable | Values |
|---|---|
| initial learning rate | .0001 |
| learning epochs | 150 |
| learning rate decay | .3 |
| learning rate decay epochs | 50, 100 |
| SGLD steps $\eta$ | 20 |
| Buffer-size | 10000 |
| reinitialization frequency $\rho$ | .05 |
| SGLD step-size $\alpha$ | 1 |
| SGLD noise $\sigma$ | .01 |

Table 4: Hyperparameters

## B   SAMPLE QUALITY EVALUTION

In this section we describe the details for reproducing the Inception Score (IS) and FID results reported in the paper. First we note that both IS and FID are scores computed based on a pretrained classifier network, and thus can be very dependent on the exact model/code repository used. For a more detailed discussion on the variability of IS, please refer to Barratt & Sharma (2018). To gauge our model against the other papers, we document our attempt to fairly compare the scores across

papers in Table 6. As a direct comparison of IS, we got 8.76 using the code provided by Du & Mordatch (2019), and is better than their best reported score of 8.3. For FID, we used the official implementation from Heusel et al. (2017). Note that FID computed from this repository assigned much worse FID than reported in Chen et al. (2019).

**Conditional vs unconditional samples.** Since we are interested in training a Hybrid model, our model, by definition, is a conditional generative model as it has access to label information. In Table 5, **unconditional** samples mean samples directly obtained from running SGLD using $p(x)$. **Conditional** samples are obtained by taking the max of our $p(y|x)$ model. The reported scores are obtained by keeping the top 10 percentile samples with the highest $p(y|x)$ values. Scores obtained on a "single" model are computed directly on the training replay buffer of the last checkpoint. "Ensemble" here are obtained by lumping together 5 buffers over the last few epochs of training. As we initialize SGLD with uniform noise, using the training buffer is exactly the same as re-sampling from the model.

| Method | Conditional | | Unconditional | |
|---|---|---|---|---|
| | single | ensemble | single | ensemble |
| JEM (Ours) | - | 8.76 | 7.82 | 7.79 |
| EBM (D&M) | 8.3 | X | 6.02 | 6.78 |

Table 5: Conditional vs. unconditional Inception Scores.

| Method | Inception Score | | | FID | | |
|---|---|---|---|---|---|---|
| | from paper | B&S | D&M | from paper | H | D&M |
| Residual Flow | X | 3.6 | - | 46.4 | - | - |
| Glow | X | - | 3.9 | 48.9* | 107 | - |
| JEM (Ours) | X | 7.13 | 8.76 | X | 38.4 | - |
| JEM $p(\mathbf{x}|y)$ factored | X | - | 6.36 | X | 61.8 | - |
| EBM (D&M) | 8.3 | - | 8.3 | 37.9 | - | 37.9 |
| SNGAN | 8.59 | - | - | 25.5 | - | - |
| NCSN | 8.91 | - | - | 25.3 | - | - |

Table 6: The headings: B&S, D&M, and H denotes scores computed using code provided by Barratt & Sharma (2018), Du & Mordatch (2019),Heusel et al. (2017). *denotes numbers copied from Chen et al. (2019), but not the original papers. As unfortunate as the case is with Inception Score and FID (i.e., taking different code repository yields vastly different results), from this table we can still see that our model performs well. Using D&M Inception Score we beat their own model, and using the official repository for FID we beat the Glow [3]model by a big margin.

---

[3]Code taken from `https://github.com/y0ast/Glow-PyTorch`

## C  FURTHER HYBRID MODEL SAMPLES

Additional samples from CIFAR10 and SVHN can be seen in Figure 7 and samples from CIFAR100 can be seen in Figure 8

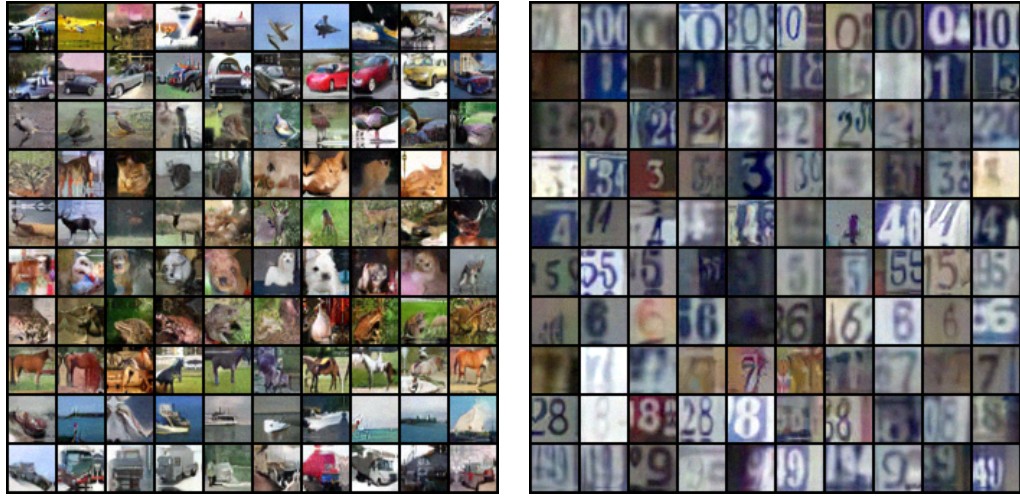

Figure 7: Class-conditional Samples. Left to right: CIFAR10, SVHN.

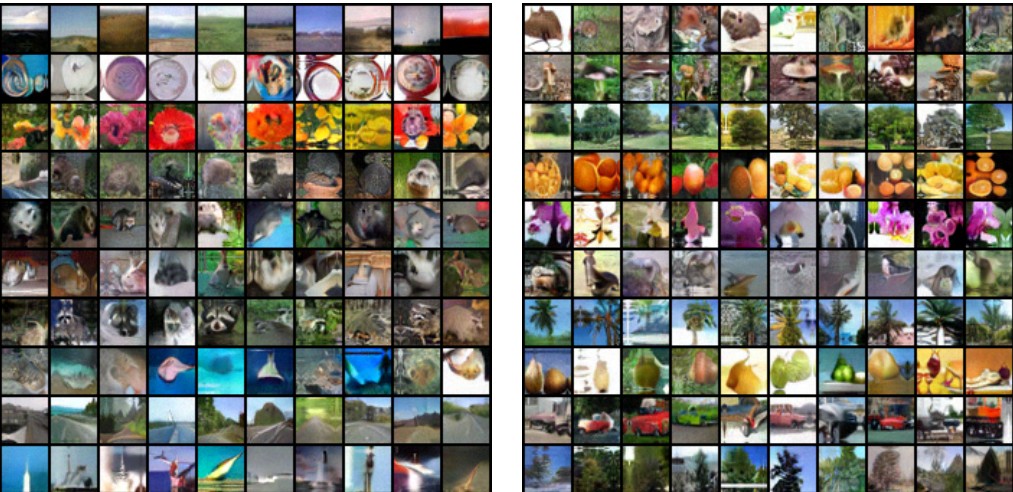

Figure 8: CIFAR100 Class-conditional Samples.

## D    QUALITATIVE ANALYSIS OF SAMPLES

Visual quality is difficult to quantify. Of the known metrics like IS and FID, using samples that have higher $p(y|\mathbf{x})$ values results in higher scores, but not necessary if we use samples with higher $\log p(\mathbf{x})$. However, this is likely because of the downfalls of the evaluation metrics themselves rather than reflecting true sample quality.

Based on our analysis (below), we find

1. Our $\log p(\mathbf{x})$ model assigns values that cluster around different means for different classes. The class automobiles has the highest $\log p(\mathbf{x})$. Of all generated samples, all top 100 samples are of this class.

2. Given the class, the samples that have higher $\log p(\mathbf{x})$ values all have white background and centered object, and lower $\log p(\mathbf{x})$ samples have colorful (e.g., forest-like) background.

3. Of all samples, higher $p(y|\mathbf{x})$ values means clearly centered objects, and lower $p(y|\mathbf{x})$ otherwise.

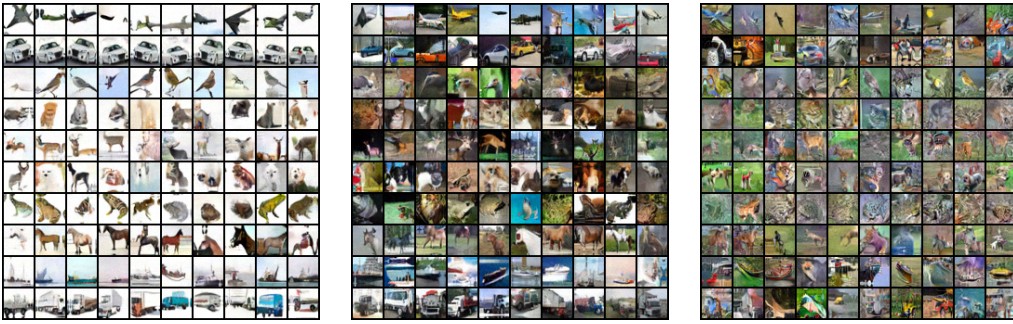

Figure 9: Each row corresponds to 1 class, subfigures corresponds to different values of $\log p(\mathbf{x})$. left: highest, mid: random, right: lowest.

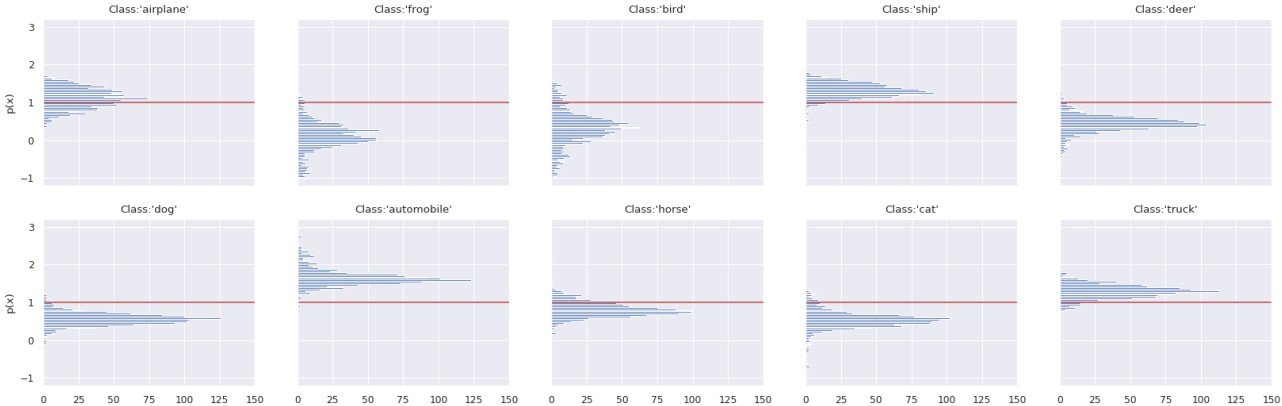

Figure 10: Histograms (oriented horizontally for easier visual alignment) of $\log p(\mathbf{x})$ arranged by class.

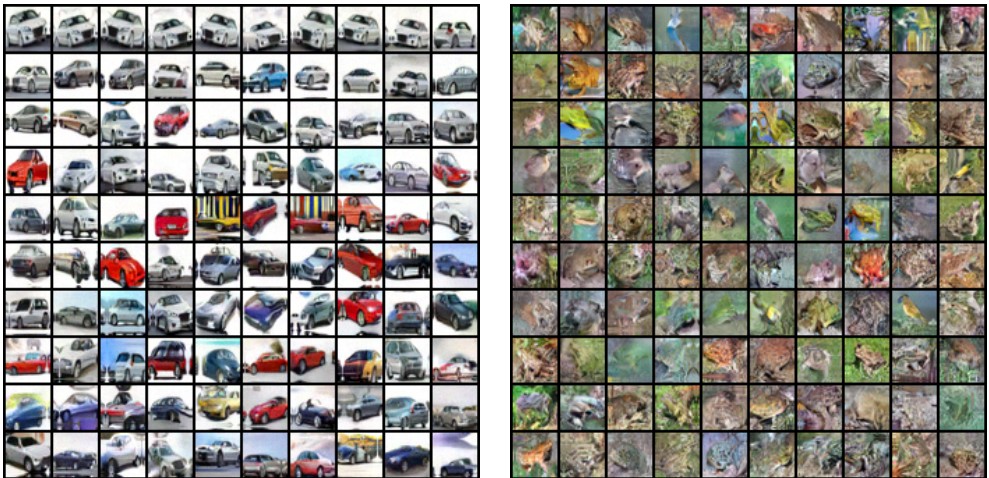

Figure 11: left: samples with highest $\log p(\mathbf{x})$, right: left: samples with lowest $\log p(\mathbf{x})$

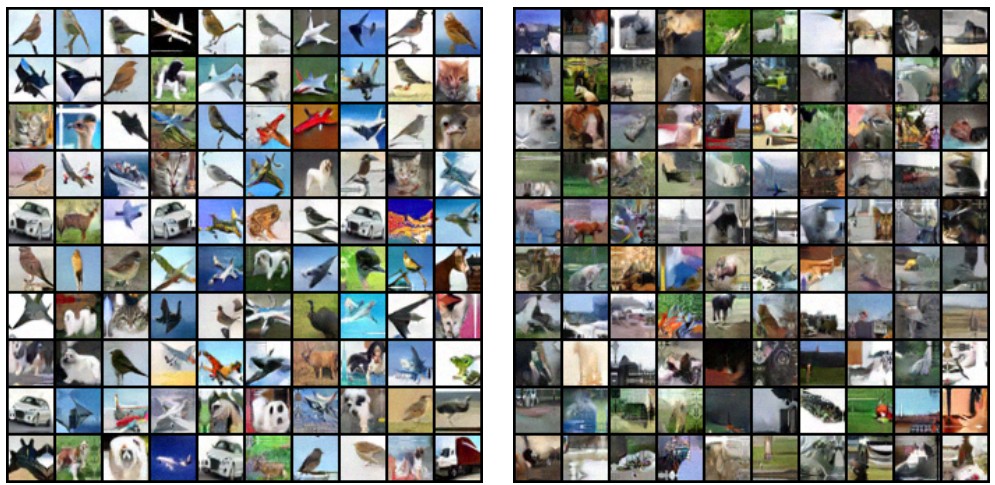

Figure 12: left: samples with highest $p(y|x)$, right: left: samples with lowest $p(y|x)$

# E  CALIBRATION

## E.1  EXPECTED CALIBRATION ERROR

Expected Calibration Error (ECE) is a metric to measure the calibration of a classifier. It works by first computing the confidence, $\max_y p(y|\mathbf{x}_i)$, for each $\mathbf{x}_i$ in some dataset. We then group the items into equally spaced buckets $\{B_m\}_{m=1}^M$ based on the classifier's output confidence. For example, if $M = 20$, then $B_0$ would represent all examples for which the classifier's confidence was between 0.0 and 0.05.

We then define:

$$\text{ECE} = \sum_{m=1}^{M} \frac{|B_m|}{n} |\text{acc}(B_m) - \text{conf}(B_m)| \tag{10}$$

where $n$ is the number of examples in the dataset, $\text{acc}(B_m)$ is the averaged accuracy of the classifier of all examples in $B_m$ and $\text{conf}(B_m)$ is the averaged confidence over all examples in $B_m$.

For a perfectly calibrated classifier, this value will be 0 for any choice of $M$. In our analysis, we choose $M = 20$ throughout.

## E.2  FURTHER RESULTS

We find that JEM also improves calibration on CIFAR10 as can be seen in Table 13. There we see an improvement in calibration, but both classifiers are well calibrated because their accuracy is so high. In a more interesting experiment, we limit the size of the training set to 4,000 labeled examples. In this setting the accuracy drops to 78.0% and 74.9% in the baseline and JEM, respectively. Given the JEM can be trained on unlabeled data, we treat the remainder of the training set as unlabeled and train in a semi-supervised manner. We find this gives a noticeable boost in the classifier's calibration as seen in Figure 13. Surprisingly this did not improve generalization. We leave exploring this phenomenon for future work.

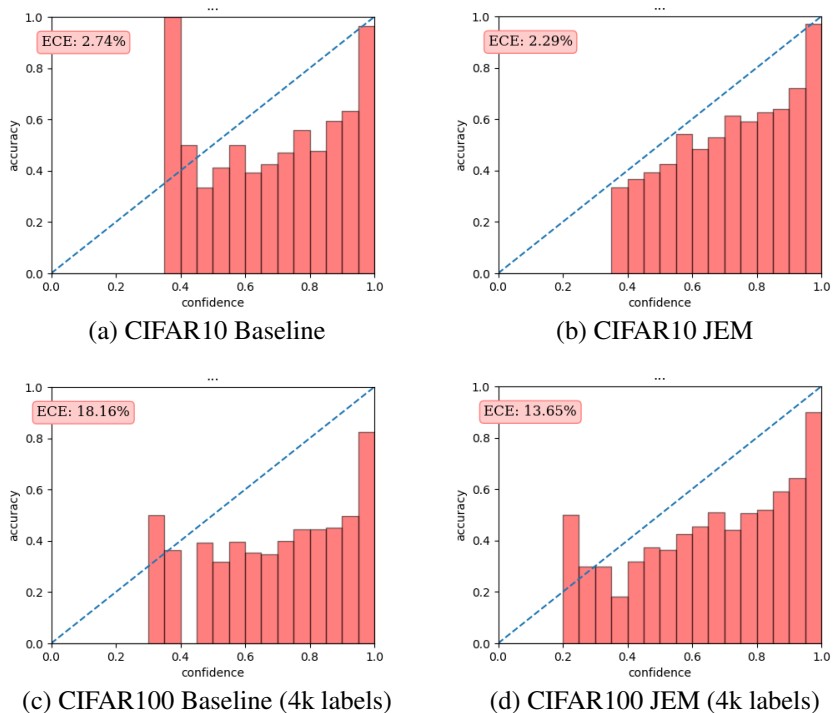

(a) CIFAR10 Baseline      (b) CIFAR10 JEM

(c) CIFAR100 Baseline (4k labels)      (d) CIFAR100 JEM (4k labels)

Figure 13: CIFAR10 Calibration results

# F OUF-OF-DISTRIBUTION DETECTION

## F.1 EXPERIMENTAL DETAILS

To obtain OOD results for unconditional Glow, we used the pre-trained model and implementation of `https://github.com/y0ast/Glow-PyTorch`. We trained a Class-Conditional model as well using this codebase which was used to generate the class-conditional OOD results.

We obtained the IGEBM of Du & Mordatch (2019) from their open-source implementation at `https://github.com/openai/ebm_code_release`. For likelihood and likelihood-gradient OOD scores we used their pre-trained `cifar10_large_model_uncond` model. We were able to replicate the likelihood based OOD results presented in their work. We implemented our likelihood-gradient approximate-mass score on top of their codebase. For predictive distribution based OOD scores we used their `cifar_cond` model which was the model used in their work to generate their robustness results.

## F.2 FURTHER RESULTS

Figure 7 contains results on two datasets, Constant and Uniform, which were omitted for space. Most models perform very well at the Uniform dataset. On the Constant dataset (all examples = 0) generative models mainly fail – with JEM being the only one whose likelihoods can be used to derive a predictive score function for OOD detection. Intrestinly, we could not obtain approximate mass scores on this dataset from the Glow models due to numerical stability issues.

| | | | | CIFAR10 | | | |
|---|---|---|---|---|---|---|---|
| Score | Model | SVHN | Uniform | Constant | Interp | CIFAR100 | CelebA |
| $\log p(\mathbf{x})$ | Unconditional Glow | .05 | 1.0 | 0.0 | .51 | .55 | .57 |
| | Glow Supervised | .07 | 1.0 | 0.0 | .45 | .51 | .53 |
| | IGEBM | .63 | 1.0 | .30 | **.70** | .50 | .70 |
| | JEM (Ours) | **.67** | 1.0 | **.51** | .65 | **.67** | **.75** |
| $\max_y p(y\|\mathbf{x})$ | WRN-baseline | **.93** | **.97** | **.99** | **.77** | .85 | .62 |
| | Class-Conditional Glow | .64 | 0.0 | .82 | .61 | .65 | .54 |
| | IGEBM | .43 | .05 | .60 | .69 | .54 | .69 |
| | JEM (Ours) | .89 | .41 | .84 | .75 | **.87** | **.79** |
| $\left\lVert \frac{\partial \log p(\mathbf{x})}{\partial \mathbf{x}} \right\rVert$ | Unconditional Glow | **.95** | .99 | NaN | .27 | .46 | .29 |
| | Class-Conditional Glow | .47 | .99 | NaN | .01 | .52 | .59 |
| | IGEBM | .84 | .99 | 0.0 | .65 | .55 | .66 |
| | JEM (Ours) | .83 | **1.0** | **.75** | **.78** | **.82** | **.79** |

Table 7: OOD Detection Results. Values are AUROC.

# G ATTACK DETAILS AND FURTHER ROBUSTNESS RESULTS

We use foolbox (Rauber et al., 2017) for our experiments. PGD uses binary search to determine minimal epsilons for every input and we plot the resulting robustness-distortion curves. PGD runs with 20 random restarts and 40 iterations. For the boundary attack, we run default foolbox settings with one important difference. The random initialization often fails for JEM and thus we initialize the attack with a correclty classified input of another class. This other class is chosen based on the top-2 prediction for the image to be attacked. As all our attacks are expensive to run, we only attacked 300 randomly chosen inputs. The same randomly chosen inputs were used to attack each model.

In Figure 14 we see the results of the boundary attack and pointwise attack on JEM and a baseline. The main point to running these attacks was to demonstrate that our model was not able to "cheat" by having vanishing gradients through our gradient-based sampling procedure. Since PGD was more successful than these gradient-free methods, this is clearly not the case and the attacker was able to use the gradients of the sampling procedure to attack our model. Further, we observe the same

behavior across all attacks; the EBM with 0 steps sampling is more robust than the baseline and the robustness increases as we add more steps of sampling.

We also compare JEM to the IGEBM of Du & Mordatch (2019) with 10 steps of sampling refinement, see Figure 15. We run the same gradient-based attacks on their model and find that despite not having competitive clean accuracy, it is quite robust to large $\epsilon$ attacks – especially with respect to the $L_\infty$ norm. After $\epsilon = 12$ their model is more robust than ours and after $\epsilon = 18$ it is more robust than the adversarial training baseline. With respect to the $L_2$ norm their model is more robust than the adversarial training baseline above $\epsilon = 280$ but remains less robust than JEM until $\epsilon = 525$.

We believe these results demonstrate that EBMs are a compelling class of models to explore for further work on building robust models.

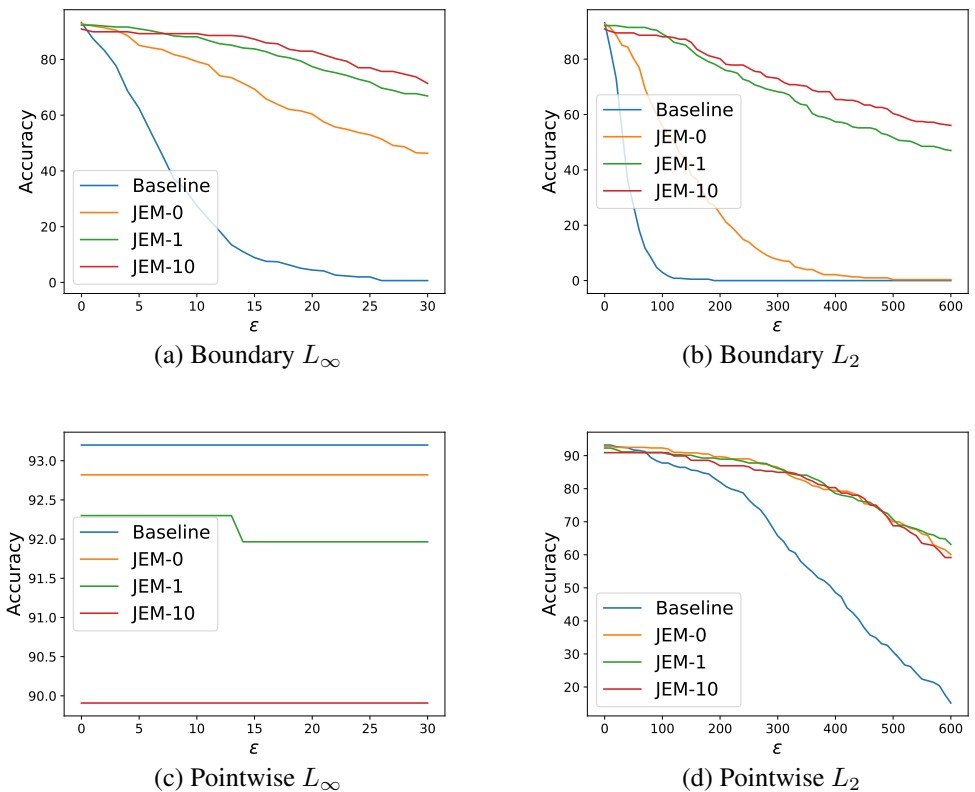

Figure 14: Gradient-free adversarial attacks.

## G.1 EXPECTATION OVER TRANSFORMATIONS

Our SGLD-based refinement procedure is stochastic in nature and it has been shown that stochastic defenses to adversarial attacks can provide a false sense of security (Athalye et al., 2018). To deal with this, when we attack our stochastically refined classifiers, we average the classifier's predictions over multiple samples of this refinement procedure. This makes the defense more deterministic and easier to attack. We redefine the logits of our classifier as:

$$\log p_n^k(y|\mathbf{x}) = \frac{1}{n}\sum_{i=1}^{n}\log p(y|\mathbf{x}_i), \qquad \mathbf{x}_i \sim \text{SGLD}(\mathbf{x}, k) \tag{11}$$

where we have defined $\text{SGLD}(\mathbf{x}, k)$ as an SGLD chain run for $k$ steps seeded at $\mathbf{x}$. Intuitively, we draw $n$ different samples $\{\mathbf{x}_i\}_{i=1}^n$ from our model seeded at input $\mathbf{x}$, then compute $\log p(y|\mathbf{x}_i)$ for each of these samples, then average the results. We then attack these averaged logits with PGD to generate the results in Figure 5. We experimented with different numbers of samples and found that

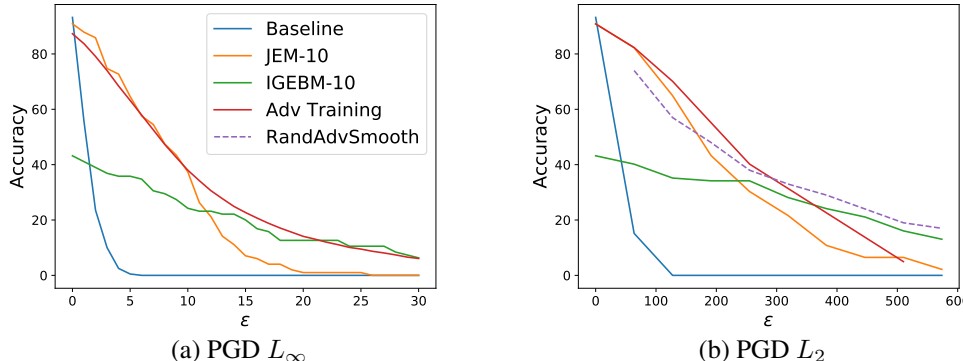

(a) PGD $L_\infty$            (b) PGD $L_2$

Figure 15: PGD attacks comparing JEM to the IG EBM of Du & Mordatch (2019).

10 samples yields very similar results to 5 samples on JEM with one refinement step (see Figure 16). Because 10 samples took very long to run on the JEM model with ten refinement steps, we settled on using 5 samples in the results reported in the main body of the paper.

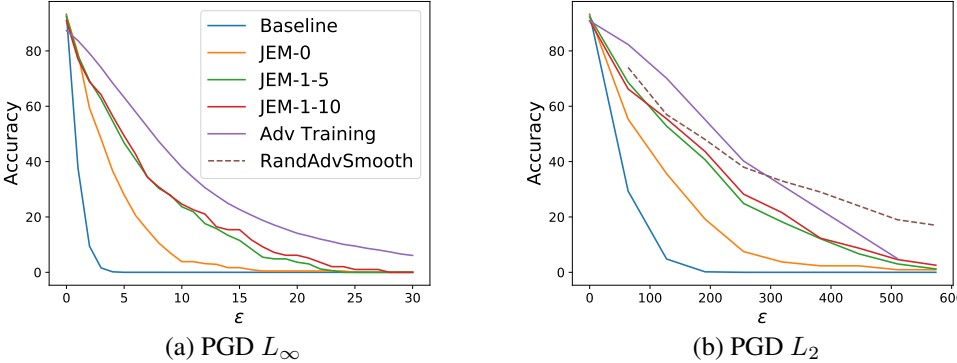

(a) PGD $L_\infty$            (b) PGD $L_2$

Figure 16: Comparing the effect of the number of samples in the EOT attack. We find negligible difference between 5 and 10 for JEM-1 (red and green curves).

## G.2 TRANSFER ATTACKS

We would like to see if JEM's refinement procedure can correct adversarial perturbed inputs – inputs which cause the model to fail. To do this, we generate a series of adversarial examples for JEM-0, with respect to the $l_\infty$ norm, and test the accuracy of JEM-$\{1,10\}$ on these examples. Ideally, with further refinement the accuracy will increase. The results of this experiment can be seen in Figure 17. We see here that JEM's refinement procedure can correct for adversarial perturbations.

## H A DISCUSSION ON SAMPLERS

### H.1 IMPROPER SGLD

Recall the transition kernel of SGLD:

$$\mathbf{x}_0 \sim p_0(\mathbf{x})$$

$$\mathbf{x}_{i+1} = \mathbf{x}_i - \frac{\alpha}{2} \frac{\partial E_\theta(\mathbf{x}_i)}{\partial \theta} + \epsilon, \qquad \epsilon \sim \mathcal{N}(0, \alpha)$$

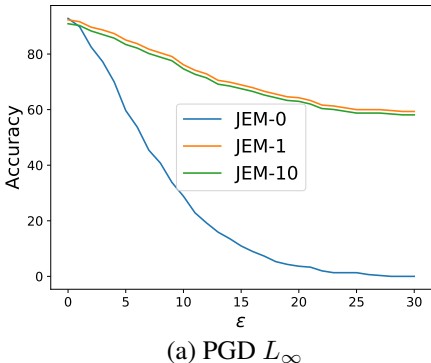

(a) PGD $L_\infty$

Figure 17: PGD transfer attack $L_\infty$. We attack JEM-0 and evaluate success of the same adversarial examples under JEM-1 and JEM-10. Whenever an adversarial example is refined back to its correct class, we set the distance to infinity. Note that the adversarial examples do not transfer well from JEM-0 to JEM-1/-10.

In the proper formulation of this sampler (Welling & Teh, 2011), the step-size and the variance of the Gaussian noise are related $\mathrm{Var}(\epsilon) = \alpha$. If the stepsize is decayed with a polynomial schedule, then samples from SGLD converge to samples from our unnomralized density as the number of steps goes to $\infty$.

In practice, we approximate these samples with a sampler that runs for a finite number of steps. When using the proper step-size to noise ratio, the signal from the gradient is overtaken by the noise when step-sizes are large enough to be informative. In practice the sampler is typically "relaxed" in that different values are used for the step-size and the amount of Guassian noise added – typically the amount of noise is significantly reduced.

While we are no longer working with a valid MCMC sampler, this approximation has been successfully applied in practice in most recent work scaling EBM training to high dimensional data (Nijkamp et al., 2019b;a; Du & Mordatch, 2019) with the exception of Song & Ermon (2019) (which develops a clever work-around). The model they train is actually an ensemble of models trained on data with different amounts of noise added. They use a proper SGLD sampler decaying the step size as they sample, moving from their high-noise models to their low-noise models. This provides one possible explanation for the compelling results of their model.

In our work we have set the step-size $\alpha = 2$ and draw $\epsilon \sim \mathcal{N}(0, .01^2)$. We have found these parameters to work well across a variety of datasets, domains, architectures, and sampling procedures (persistent vs. short-run). We believe they are a decent "starting place" for energy-functions parameterized by deep neural networks.

## H.2 PERSISTENT OR SHORT-RUN CHAINS?

Both persistent and short-run markov chains have been able to succesfully train EBMs. Nijkamp et al. (2019a) presents a careful study of various samplers which can be used and the tradeoffs one makes when choosing one sampler over another. In our work we have found that if computation allows, short-run MCMC chains are preferable in terms of training stability. Given that each step of SGLD requires approximately the computation of 1 training iteration of a standard classifier we are incentivized to find a sampler which can stably train EBMs requiring as few steps as possible per training iteration.

In our experiments we found the smallest number of SGLD steps we could take to stably train an EBM at the scale of this work was 80 steps. Even so, these models eventually would diverge late into training. At 80 steps, we found the cost of training to be prohibitively high compared to a standard classifier.

We found that by using persistent markov chains, we could further reduce the number of steps per iteration to 20 and still allow for relatively stable training. This gave a 4x speedup over our fastest short-run MCMC sampler. Still, this PCD sampler was noticebly less stable than the fastest short-run sampler we could use but we found the multiple factor increase in speed to be a worth-while trade-off.

If time allows, we recommend using a short-run MCMC sampler with a large enough number of steps to be stable. Given that is not always possible on problems of scale, PCD can be made to work more efficiently, but at the cost of a greater number of stability-related hyper-parameters. These additional parameters include the buffer size and the re-initialization frequency of the Markov chains. We found both to be important for training stability and found no general recipe for which to set them. We ran most of our experiments with re-initialization frequency at $5\%$.

A particualrly interesting observation we discovered while using PCD is that the model would use the length of the markov chains to encode semantic information. We found that when training models on CIFAR10, when chains were young they almost always could be identified as frogs. When chains were old they could almost always be identified as cars. This behavior is likely some degeneracy of PCD which would not be possible with a short-run MCMC since all chains have the same length.

## H.3  DEALING WITH INSTABILITY

Training a model with the gradient estimator of Eq. (2) can be quite unstable – especially when combined with other objective as was the case with all models presented in this work. There exists a "stable region" of sorts when training these models where the energy values of the true data are in the same range as the energy values of the generated samples. Intuitively, if the generated samples create energies that are not trivially separated from the training data, then real learning has to take place. Nijkamp et al. (2019b;a) provide a careful analysis of this and we refer the reader there for a more in-depth analysis.

We find that when using PCD occasionally throughout training a sample will be drawn from the replay buffer that has a considerably higher-than average energy (higher than the energy of a random initialization). This causes the gradients w.r.t this example to be orders of magnitude larger than gradients w.r.t the rest of the examples and causes the model to diverge. We tried a number of heuristic approaches such as gradient clipping, energy clipping, ignoring examples with atypical energy values, and many others but could not find an approach that stabilized training and did not hurt generative and discriminative performance.

The only two approaches we found to consistently work to increase stability of a model which has diverged is to 1) decrease the learning rate and 2) increase the number of SGLD steps in each PCD iteration. Unfortunately, both of these approaches slow down learning. We also had some success simply restarting models from a saved checkpoint with a different random seed. This was the main approach taken unless the model was late into training. In this case, random restarts were less effective and we increased the number of SGLD steps from 20 to 40 which stabilized training.

While we are very optimistic about the future of large-scale EBMs we believe these are the most important issues that must be addressed in order for these models to be succeful.

