# OpenReview forum: "Your classifier is secretly an energy based model and you should treat it like one"
_ICLR.cc/2020/Conference — Accept (Talk)_

### Official Review · AnonReviewer2 · 2019-10-09
**Official Blind Review #2**

**Rating:** 8

**Review:**

This work is an attempt to bridge the gap between discriminative models, which currently obtain the state of the art on most classification problems, and generative models, which (through a model of the marginal p(x)) have the potential to shine on many tasks beyond generalization to a hold-out set with minimal shift in distributions: out of distribution detection, better generalization out of distribution, unsupervised learning etc.

While much of the current work is related to normalizing flows / invertible neural networks, the authors here propose a quite simple but appealing method: A standard neural classifier is taken and the softmax is layer chopped off and replaced by an energy based model, which models the joint probability p(x,y) instead of the posterior p(y|x). The advantage is an additional degree of freedom in the scale of the logit vector, which is would have been otherwise normalized by the softmax layer and now can now model the data distribution. The downside is the loss in ease of training. Whereas (discriminative) deep networks can be easily trained by gradient descent on a cross-entropy objective, the partition function in the energy model makes this un tractable. This is addressed through sampling, similar to (Welling & Teh, 2011).

One of the biggest achievements reported by the authors is that the performance on discriminative tasks is not hurt (much) by adding the generative model. There is only a 3 point gap between Wide-ResNet and the proposed model (92.9% vs. 95.8%) … but on what dataset? 3 datasets are mentioned in the experimental section, but table 1 does not mention on which datasets the accuracy is reported. My guess is that this is a mean or mixture, since GEM performances of 96.7% and 72.2% are reported for SVHN and CIFAR10, respectively, but this should be made clearer.

On out of distribution detection, could the authors comment on the histograms in table 2, in particular the difference between the new measure (AM JEM) compared to JEM log p(x) on CelebA? The proposed measure does not seem to fare well here.

Although the method does not outperform the gold standard of adversarial training, I found the models robustness to adversarial examples quite appealing, given that it was not trained for this objective (which also means that it does not require an adaptation to a norm).

I was very impressed by Figure 6 showing distal adversarial initialized from random images, showing pretty clear images of the modelled class. The modelled variations require more investigation to verify whether we have a collapse for each class, but the results look very promising.

The paper is well written and easy to understand. A couple of details on the training procedure are missing in the experimental part. It is stated that, both, p(y|x) and the generative part p(x), are optimized, but how are these exactly integrated? Given the difficult in training this model reported in the paper, this seems to be particularly important.

I also appreciated the description of the limitations of the algorithm, and the details in the appendix (ICLR should go back to unlimited paper lengths, btw.).

More information on complexity (training times etc.) should also be helpful.


**Experience Assessment:**

I have read many papers in this area.

**Review Assessment: Checking Correctness Of Derivations And Theory:**

I carefully checked the derivations and theory.

**Review Assessment: Checking Correctness Of Experiments:**

I carefully checked the experiments.

**Review Assessment: Thoroughness In Paper Reading:**

I read the paper thoroughly.

---

> ### Author Response · Authors · 2019-11-08
> **Response to R2**
>
> We thank you for your time reviewing our work.
>
> You are correct, the results presented in table 1 are CIFAR10. We forgot to add this to the caption. We have updated the caption to fix this. Results on other datasets can be found in the text of Section 5.1.
>
> Regarding the performance of our approximate-mass measure for OOD detection on CelebA, we refer you to Table 3, bottom row, right-most column. This metric gets AUROC = .79 on this dataset, higher actually than unnormalized logp(x) which gets AUROC = .75. The metric may appear to perform worse than the likelihood in the histogram but the low-valued tails are larger and thus the AUROC is higher.
>
> Regarding the training procedure, we optimize a training objective which is log p(y|x) + log p(x). This is equivalent to optimizing log p(x, y). These two terms are combined by adding their gradients exactly. This is equivalent to an equal weighting of the two terms. We choose this way to factor the learning objective since p(y|x) is a normalized distribution and we can train with maximum likelihood exactly, avoiding a biased and tricky gradient estimation problem. We are aware that other hybrid models typically downweight the log p(x) term. We do not do that here.
>
> Regarding the training time and hardware requirements, we note that all models were trained on a single GPU in approximately 36 hours. We have added a few sentences to section 5 in the text to clarify this.

---

> > ### Comment · AnonReviewer2 · 2019-11-14
> > **Post rebuttal**
> >
> > Thank you for the responses, which addressed my questions.

---

### Official Review · AnonReviewer3 · 2019-10-23
**Official Blind Review #3**

**Rating:** 8

**Review:**

The paper uses energy-based model interpretation for the logits of standard discriminative neural network models to define a generative model inside a classifier that proves useful in many downstream tasks such as uncertainty quantification, out-of-distribution detection, etc.
Although there has been previous work attempting to bridge discriminative classifiers with generative modeling, this work proves to be competitive with both specialized models on discriminative/generative tasks as well as in many downstream tasks such as out-of-distribution detection, calibration, and adversarial robustness. The paper provides a clear exposition of the method, succeeds to discuss related work it bases on, conducts a thorough experimental study providing convincing explanations for results and does not hide the limitations of the work (high computational requirements, optimization difficulties connected with training energy-based model and the method used, limited approximation of the true energy). Overall, the paper provides a substantial contribution and paves the way for further work improving this joint discriminative - generative setting. However, there are points I would like the paper to address for better exposition.
1. It would benefit the paper showing that samples with higher unnormalized likelihood are visually more compelling than those with lower likelihood.
2. On CIFAR100 the accuracy drop from the reference value is larger than for datasets with 10 classes, could it be due the logits dimension is higher and challenges optimization?
3. It would also be helpful to clarify whether application of the proposed method is primarily restricted by the computational complexity or is there any property inherent to energy-based models that makes treating high-dimensional data challenging?

Minor remark
- Although the paper doesn't state on which dataset results shown in Table 1 were obtained, I suspect its CIFAR10, please specify this.

**Experience Assessment:**

I have read many papers in this area.

**Review Assessment: Checking Correctness Of Derivations And Theory:**

I assessed the sensibility of the derivations and theory.

**Review Assessment: Checking Correctness Of Experiments:**

I carefully checked the experiments.

**Review Assessment: Thoroughness In Paper Reading:**

I read the paper thoroughly.

---

> ### Author Response · Authors · 2019-11-08
> **Response to R3**
>
> We thank you for your time reviewing our work. We will address your concerns in order:
>
> 1) Visual quality is difficult to quantify. Of the known metrics like IS and FID, using samples that have higher p(y|x) values results in higher scores, but not necessary if we use samples with higher p(x).  However, this is likely because of the downfalls of the evaluation metrics themselves rather than reflecting true sample quality.
>
> Based on our analysis of CIFAR10 (below), we find
>     -Our p(x) model assigns values that cluster around different means for different classes.  The class automobiles has the highest p(x).  Of all generated samples, all top 100 samples are of this class.
>     -Given the class, the samples that have higher p(x) values all have white background and centered object, and lower p(x) samples have colorful (e.g., forest-like) background.
>     -Of all samples, higher p(y|x) values means clearly centered objects, and lower p(y|x) otherwise.
>
> We completely agree with you that adding these analyese will strengthen the paper, and we have added this discussion with their corresponding images in the revised appendix.
>
>
> 2) On CIFAR10 we see our accuracy drop from 95.2% to 92.9% and on CIFAR100 we see accuracy drop from 74.2% to 72.2%. This is a 2-3% drop on both datasets. These numbers are from the exact same model with and without JEM training. In these settings the decrease in accuracy is relatively consistent.
>
> Perhaps you are referring to the accuracy of our JEM models compared to state-of-the-art discriminative classifiers on these datasets? Yes, in this setting we have a 2-3% drop from the best wide-resnet classifier with all forms of regularization added. On CIFAR100 we have approximately a 8% drop compared to the best wide-resnet.
>
> Our best guess to explain this phenomenon is that competitive accuracy on CIFAR100 is much lower than competitive accuracy on CIFAR10 meaning that much more overfitting is happening on CIFAR100 than CIFAR10 (since all models achieve a training accuracy of 100% at the end of training).
>
> In our JEM models we remove two important forms of regularization, batch norm and dropout, which we found to have negligible impact on CIFAR10 but less negligible impact on CIFAR100. This is backed up by the fact that our baseline classifier with these regularizers removed achieves 74.4% accuracy, closer to that of our JEM model.
>
> We feel that the removal of these regularizers provides an explanation for the decrease in relative performance.
>
>
> 3) This is an interesting point. We are very excited about the future of EBMs and we are generally of the belief that the application of EBMs is currently limited by the fragility of the tools we use to train them. So yes, we do believe if one had access to considerable computational resources then one should be able scale these methods presented to larger datasets, but we do believe there would be considerable engineering cost in doing so.
>
> We feel the most useful next steps to work on in the EBM-space are more stable and efficient training objectives which will increase the scale of problems to which we can apply these methods.
>
>
> Minor Remark) Yes you are correct that table presents CIFAR10 results and we did indeed forget to label it as such. This has been changed in our revised version.

---

### Official Review · AnonReviewer4 · 2019-10-31
**Official Blind Review #4**

**Rating:** 6

**Review:**

This paper introduces the idea of energy based model to the traditional classifier, and proposes a new framework to improve the performances of the model in multiple aspects. The idea of reinterpreting the traditional classifier is very interesting, and the experiments show some good results of the proposed method.

Here are my main concerns of the current paper:
1. The training procedure seems to be very sensitive, and the SGLD may take a long time at each iteration to converge. This may be a big limitation of the proposed method.
2. According to equation (8), the proposed method is having a trade-off between classification and generation, and this seems to be the key to improve the performance of the model in generation by sacrificing some classification accuracy. I think author should emphasize this instead of energy based model.
3. The presentation is not very clear in section 5. What is the task of calibration, and what is the definition of ECE?
4. The robustness guarantee seems too good to be true. Although the authors claim that they allow the attacker to have access to the gradient  of SGLD, the SGLD will add noise during the forward process, this will obfuscate the gradient. In this sense, I don’t think the proposed method will have the strong robustness as they claimed.

----------------
Post-Rebuttal Comments:
Thanks for addressing my concerns. Although I think the proposed method is not comprehensive to check obfuscated gradients, I do think the current version is a good fit for ICLR, and I decide to increase my score.

**Experience Assessment:**

I have read many papers in this area.

**Review Assessment: Checking Correctness Of Derivations And Theory:**

I carefully checked the derivations and theory.

**Review Assessment: Checking Correctness Of Experiments:**

I assessed the sensibility of the experiments.

**Review Assessment: Thoroughness In Paper Reading:**

I read the paper at least twice and used my best judgement in assessing the paper.

---

> ### Author Response · Authors · 2019-11-08
> **Response to R4**
>
> (PART 1 OF 2)
>
> We thank you for your time reviewing our work. We will address your concerns in order and we have updated the manuscript accordingly. We hope these changes will encourage you to change your score:
>
> 1) Your concerns on the sensitivity and speed of SGLD training of EBMs.
>
> Regarding your concern about sensitivity:
>
> While we agree that SGLD training of EBMs can be sensitive to hyper-parameter settings, we note that throughout our work we used the exact same hyper-parameters for every model and every dataset. We also found these settings transferred well to datasets such as MNIST which we did not present in our paper. Further, we found these same settings worked well across a variety of model architectures such as MLPs, non-resnet convnets, and resnets. This was stated in Appendix  G.2 of our paper, but we have added it to the main body of our paper for clarity. This hyper-parameter transferability behavior has also been reported in prior work on EBM training such as [1, 2].
>
> Regarding your other concern about the convergence time:
>
> In our work we put a great deal of focus into being able to train as quickly as possible with minimal hardware requirements. We have been able to train EBMs with far fewer SGLD steps per training iteration than in previous work and found that at these settings stable training can still take place. All of our models were trained on a single GPU and each training run took $<36$ hours. While this is slower than training a standard classifier on these datasets, our training speed falls comfortably in the range of other popular classes of generative models such as flows [3] and GANs [4].
>
> We admit that we did not put enough emphasis on these two facts in our original draft and we have added this information in section 5. We hope this clarifies your concerns regarding training sensitivity and run-time.
>
> Overall we feel that training and sampling are the biggest challenges when working with EBMs. Developing improved methods for this is important further work but we also feel it is outside of the scope of our current work. The main point of our work was to demonstrate that despite the challenges which currently exist in training EBMs, they can be used to achieve a very interesting and diverse set of results on problems which other classes of generative models have not been able to achieve at this scale. These results provide a strong motivation for more work in the space of EBM training methods.
>
>
> 2) We are slightly unsure of what you mean with this point. We train our model using the factorized likelihood of Equation (8). As we explain in the following sentence, this was done to reduce bias in our training procedure, not because there is a need to weight these terms differently. We are aware that this is common practice in other hybrid-models [4, 5], but we do not do this in our model. Each term in this objective is weighted equally. While different results could possibly be achieved if we did weight each term in (8) we feel that our model's ability to weight the terms equally and still perform well at both tasks is actually a benefit of our approach over competing methods. We hope this clarifies your concerns.
>
> (CONTINUED BELOW)
>
> [1] "Implicit Generation and Generalization in Energy-Based Models"  Yilun Du, Igor Mordatch. https://arxiv.org/abs/1903.08689
> [2] "On the Anatomy of MCMC-based Maximum Likelihood Learning of Energy-Based Models"  Erik Nijkamp, Mitch Hill, Tian Han, Song-Chun Zhu, Ying Nian Wu. https://arxiv.org/abs/1903.12370
> [3] "Large Scale GAN Training for High Fidelity Natural Image Synthesis"  Andrew Brock, Jess Donahue, Karen Simonyan. https://arxiv.org/abs/1809.11096
> [4] "Glow: Generative Flow with Invertible 1x1 Convolutions"  Diederik P. Kingma, Prafulla Dhariwal. https://arxiv.org/abs/1807.03039
> [5] "Residual Flows for Invertible Generative Modeling"  Ricky T. Q. Chen, Jens Behrmann, David Duvenaud, Jörn-Henrik Jacobsen. https://arxiv.org/abs/1906.02735
> [6] "On Calibration of Modern Neural Networks"  Chuan Guo, Geoff Pleiss, Yu Sun, Kilian Q. Weinberger. https://arxiv.org/abs/1706.04599

---

> > ### Author Response · Authors · 2019-11-08
> > **Response to R4 (PART 2)**
> >
> > (PART 2 of 2)
> >
> > 3) A classifier is calibrated if the confidence of its predictive distribution p(y|x) is equivalent to its misclassification rate. We state this in plain text in the first paragraph of section 5.2. ECE is the "``Expected Calibration Error" which is a metric proposed in [6] to measure calibration of classifiers. This is clearly stated in Figure 4 and there we also provide a reference to the work which introduces this metric. To make this more clear, we have added a more formal definition of calibration to section 5.2 and have added a description of the ECE measure to the Appendix so interested readers do not have to read the referenced work to understand our analysis. We hope this makes section 5 easier to follow.
> >
> > 4) Regarding the randomness in the refinement procedure, we note that JEM-0 provides considerable robustness compared to a baseline classifier (as can be seen in Figure 5). This model is completely deterministic as we do not use SGLD to refine the inputs. So, the worst-case robustness our approach adds over a baseline is still quite considerable and holds for both the L-inf and L-2 norms.
> >
> > However, following the suggestion from your and Jeremy Cohen's comment, we have run the EOT attack which averages the model's gradients over multiple samples of the randomized defense. We do find that the robustness results from JEM-1 and JEM-10 are slightly weaker than we had initially believed, but are still a considerable improvement over JEM-0 and competitive with approaches specifically targeting norm-bounded robustness. The improved robustness from the refinement procedure does appear to improve robustness with respect to both the L-inf and the L-2 norms -- a feature most robustness-specific approaches lack. These new results can be seen in a Table 5 in our revised paper.
> >
> > We hope we have thoroughly addressed your concerns and you will choose to improve your score.
> >
> > [1] "Implicit Generation and Generalization in Energy-Based Models"  Yilun Du, Igor Mordatch. https://arxiv.org/abs/1903.08689
> > [2] "On the Anatomy of MCMC-based Maximum Likelihood Learning of Energy-Based Models"  Erik Nijkamp, Mitch Hill, Tian Han, Song-Chun Zhu, Ying Nian Wu. https://arxiv.org/abs/1903.12370
> > [3] "Large Scale GAN Training for High Fidelity Natural Image Synthesis"  Andrew Brock, Jess Donahue, Karen Simonyan. https://arxiv.org/abs/1809.11096
> > [4] "Glow: Generative Flow with Invertible 1x1 Convolutions"  Diederik P. Kingma, Prafulla Dhariwal. https://arxiv.org/abs/1807.03039
> > [5] "Residual Flows for Invertible Generative Modeling"  Ricky T. Q. Chen, Jens Behrmann, David Duvenaud, Jörn-Henrik Jacobsen. https://arxiv.org/abs/1906.02735
> > [6] "On Calibration of Modern Neural Networks"  Chuan Guo, Geoff Pleiss, Yu Sun, Kilian Q. Weinberger. https://arxiv.org/abs/1706.04599

---

### Public Comment · ~Jeremy_Cohen1 · 2019-10-02
**interesting paper**

This is a fascinating paper!   I'm impressed by the finding that JEMs match the adversarial robustness of adversarial training.   If this result holds up, it will be a Very Big Deal in the literature on adversarial robustness, since adversarial training (and every other effective adversarial defense) is tied to a particular norm, whereas JEM is not.   It is precisely because this result is so exciting that I am a little bit skeptical.   Since the proposed defense involves test-time randomization (the Gaussian noise in SGLD), the best practice when attacking is to average the input gradient over several random samples -- see the "Expectation over Transformation" (EOT) technique in [1].  Several randomized defenses previously thought effective were completely broken in [1] using EOT.  Therefore, I think it would strengthen the paper's claim if you evaluated the proposed defense against an EOT adversarial attack.

Another question I have is: how many noise samples were used when evaluating each prediction of RandAdvSmooth?

[1] "Obfuscated gradients give a false sense of security: circumventing defenses to adversarial examples."  Anish Athalye, Nicholas Carlini, and David Wagner.  ICML 2018.  https://arxiv.org/abs/1802.00420

---

> ### Author Response · Authors · 2019-10-24
> **running these results now**
>
> Jeremy,
>
> Thanks for your comments and for brining this to our attention. We agree that averaging over multiple samples will give a more reliable measure of our model's robustness. We are currently generating these results but they are quite computationally challenging to generate. Generating our initial 10-step refinement results took over a week running on many GPUs and averaging the same attack over multiple samples will take weeks to generate. We hope to have the final results completed by rebuttal time.

---

> ### Author Response · Authors · 2019-11-08
> **Following up + New Results**
>
> Jeremy,
> Thank you very much for your comments. Following your recommendation we ran an EOT version of the PGD attack by averaging over multiple samples from our refinement procedure. Following this new analysis, we find that EOT indeed slightly reduces robustness. With respect to both norms our model's robustness now falls slightly below adversarial training. However, our overall claims still hold true as we do still notice a considerable improvement from the refinement procedure over the baseline (See the updated Figure 5). We have added an explanation of the EOT procedure to Appendix G.1 and updated the plots in the main body of the manuscript.
> Thanks again!

---

### Public Comment · ~Yilun_Du2 · 2019-10-24
**Interesting Paper**

This is a interesting paper! Combining an EBM with a classifier network is an interesting idea.

I had a couple concerns about connection to [1] on out of distribution evaluation of EBMs. In [1], we show in section 4.4, we first propose to evaluate the out of distribution evaluation of EBMs, using the likelihood, and show that it performs significantly better than other likelihood models, evaluated using the exact same AUROC metrics proposed in the paper.  I hope the text can be revised to reference this earlier work.

[1] "Implicit Generation and Generalization in Energy-Based Models"  Yilun Du, Igor Mordatch.  NeurIPS 2019.  https://arxiv.org/abs/1903.08689

---

> ### Author Response · Authors · 2019-10-24
> **out-of-distribution detection**
>
> Thank you for your comment and your interest in our work.
>
> I would first like to note that we reference your work many times throughout our paper and directly compare against your model on every OOD detection metric we report. Your paper is IGEBM in our table 3. If this was not clear then we can make it more so.
>
> If your concern is simply that we do not mention that your work was first to notice that an EBM's unnormalized likelihood can perform better at OOD detection than exact likelihood models, then we are happy to add a sentence saying so in section 5.3.1.
>
> I hope this addresses your concerns.

---

> > ### Public Comment · ~Yilun_Du2 · 2019-11-06
> > **out-of-distribution detection**
> >
> > Thanks, adding a sentence in 5.3.1 would address my concerns! This is a very interesting paper!

---

### Public Comment · ~Zhijian_Ou1 · 2019-11-02
**Missing related work**

Nice to read this paper. However, the main model (as presented in Section 3) is similar to (almost the same as) the work presented in the previous work (Section 3.4 there), which has been applied for semi-supervised learning and anomaly detection (similar to Out-Of-Distribution Detection).

Yunfu Song, Zhijian Ou. Learning Neural Random Fields with Inclusive Auxiliary Generators. arxiv 1806.00271, 2018.
https://arxiv.org/abs/1806.00271

It would be nice if the authors could connect and compare to this previous work.

---

> ### Author Response · Authors · 2019-11-02
> **thanks**
>
> Zhijian,
>
> Thank you for brining your work to our attention. We will happily add a citation to your paper in our related work section.

---

### Author Response · Authors · 2019-11-08
**Post-Review Revisions**

We thank the reviewers for their thoughtful and valuable comments. We have heard your feedback and have made several minor revisions to our paper which we summarize here. We feel the paper is greatly improved after incorporating their feedback.

-As pointed out by R2 and R3, we have changed the caption in Figure 1 to indicate that these results are on CIFAR10

-Responding to R3, we have added a section into the Appendix, "Qualitative Analysis of Samples". We find these new results particularly interesting!

-Responding to R4's concerns about sensitivity and speed, we have moved some text from the Appendix to the main body about the run-time and sensitivity to hyper-parameters.

-Responding to R4, in section 5.2, we provide a more technical description of calibration and added a Section, "Calibration", to the Appendix.

-Responding to R4 and Jeremy Cohen about the stochasticity in our model and its robustness to adversarial examples, we have thoroughly re-evaluated our models using the Expectation Over Transformations attack [1] which makes randomized defenses like ours easier to attack. This new analysis finds that our models are slightly less robust than we have initially believed, but still competitive with robustness-specific approaches. We have updated Figure 5 with the new results and modified our discussion to accommodate these new results. We have also expanded the adversarial attack section in our Appendix to explain this new attack and how it was run.

[1] "Obfuscated gradients give a false sense of security: circumventing defenses to adversarial examples."  Anish Athalye, Nicholas Carlini, and David Wagner.  ICML 2018.  https://arxiv.org/abs/1802.00420

---

### Public Comment · ~Kwonjoon_Lee2 · 2019-12-21
**Related Work**

Congratulations on the ICLR oral acceptance.

We have works ([1, 2]) that advocate making neural networks simultaneously generative and discriminative (by learning energy-based model).

In [1,2], a single CNN model has shown competitive results for image synthesis, image classification, and robustness against adversarial attacks.

[1] Kwonjoon Lee, Weijian Xu, Fan Fan, and Zhuowen Tu, "Wasserstein Introspective Neural Networks", CVPR 2018.
[2] Long Jin, Justin Lazarow, and Zhuowen Tu, "Introspective Classification with Convolutional Nets", NeurIPS 2017.

---

> ### Author Response · Authors · 2019-12-21
> **thanks**
>
> Thank you for brining your work to our attention. Shortly after releasing our work someone else brought these papers to our attention. We agree it is indeed related and are planning to add a reference in our related work section in the camera-ready version of our paper.

---

> > ### Public Comment · ~Kwonjoon_Lee1 · 2020-02-12
> > **Thanks**
> >
> > I am looking forward to a discussion about the methods in the paper.

---

### Public Comment · ~Kimin_Lee1 · 2019-12-22
**Related work**

Dear authors,

Thank you for the interesting paper and congratulations on the acceptance at ICLR.

Combining an EBM with a classifier is very interesting. I would like to draw your attention to some of our previous work [1] related to this topic. Our work [1] can also be interpreted as fitting a specific form of energy function from a standard deep neural network classifier. In our work, we proposed to induce such EBM from pre-trained classifiers, rather than training from scratch. We showed that the derived classifier doesn't suffer performance degradation in classification, and also found that this kind of model can be very effective for detecting out-of-distribution and adversarial samples.

[1] Lee, Kimin, Lee, Kibok, Lee, Honglak. and Shin, Jinwoo, A simple unified framework for detecting out-of-distribution samples and adversarial attacks. In Advances in Neural Information Processing Systems, 2018. https://arxiv.org/abs/1807.03888

Thank you very much.

---

> ### Author Response · Authors · 2019-12-23
> **thanks**
>
> Thanks for your kind words about our work. Regarding the work of your own that you bring up; we feel it is an interesting paper with strong results but not very related to our own. There are a number of works which detect OOD examples by training auxiliary generative models on top of network activations. Our method works quite differently so we did not compare with any such methods. Further, while your approach trains a generative model on top of the features of a trained classifier, this is not a generative model for p(x) and data cannot be sampled from it.
>
> We thank you for your interest in our work but we do not feel your work is related enough to warrant changing our paper.

---

### Comment · AnonReviewer1 · 2019-12-23
**Classifier based energy models in the absence of labels**

Congrats on the acceptance! Very inspiring work.

I am curious about what the authors think about a line of recent work using classifiers to "boost" the performance of a base generative model.

1. Boosted Generative Models.  AAAI 2018.
2. Discriminator rejection sampling. ICLR 2019.
3. Metropolis-hastings generative adversarial networks. ICML 2019.
4. Bias Correction of Learned Generative Models using Likelihood-Free Importance Weighting. NeurIPS 2019.

Both the current work and the above line of work exploit the inductive bias of classifiers for specifying an energy function (see Section 4 of Ref. 4 above for a concise explanation). The training of the induced energy function here is multi-step (first a generative model and then a classifier) but the labels for the classifier training are artificially induced (real vs. fake) which makes the setup more broadly applicable to even unlabelled datasets.

I'd be interested to hear if the authors have additional insights to share on similarities, differences, potential hybrids etc. of their work with the above line of work. Thanks!

---

### Public Comment · ~Jianwen_Xie1 · 2020-01-03
**Related work about EBMs parameterized by deep neural networks**

Dear authors,

Congratulation on your accepted paper !!

I would like to share you some papers about training large-scale EBMs on high-dimensional data, parameterized by deep neural networks.  [1] and [2] proposed deep EBM using Spatial-Temporal ConvNet as the energy function for video generation and recovery.  [3] proposed deep EBM using volumetric ConvNet as the energy function for 3D shape patterns generation and classification.  [4] proposed multi-grid MCMC to learn EBM with ConvNet as energy function.

Thank you :)


[1] Synthesizing Dynamic Pattern by Spatial-Temporal Generative ConvNet
Jianwen Xie, Song-Chun Zhu, Ying Nian Wu (CVPR 2017)

[2] Learning Energy-based Spatial-Temporal Generative ConvNet for Dynamic Patterns
Jianwen Xie, Song-Chun Zhu, Ying Nian Wu
IEEE Transactions on Pattern Analysis and Machine Intelligence (TPAMI) 2019

[3] Learning Descriptor Networks for 3D Shape Synthesis and Analysis
Jianwen Xie *, Zilong Zheng *, Ruiqi Gao, Wenguan Wang, Song-Chun Zhu, Ying Nian Wu (CVPR) 2018

[4]  Learning generative ConvNets via multigrid modeling and sampling.
R Gao*, Y Lu*, J Zhou, SC Zhu, and YN Wu (CVPR 2018).

---

### Public Comment · ~Anh_Totti_Nguyen1 · 2020-03-30
**Related work of EMBs using classifiers**

Dear authors,

Congratulations on the Oral paper at ICLR 2020! Really nice paper indeed!! :-)

Related to your work, in PPGNs [1], we combined a p(x) (i.e. parameterized by a Denoising Autoencoder) and a p(y|x) convnet classifier to form an energy model for the joint p(x,y).
The flexibility of EBMs enabled us to plug in different p(y|x) models and form a new EBM (thus, the plug-and-play).

Your thoughts are greatly welcomed and appreciated :-)

cheers,

Anh

[1] Plug & Play Generative Networks: Conditional Iterative Generation of Images in Latent Space Nguyen et al. CVPR 2017
 https://arxiv.org/abs/1612.00005

---

### Decision · Program_Chairs · 2019-12-19

**Decision:**

Accept (Talk)

**Comment:**

This paper uses energy based model to interpret standard discriminative classifier and demonstrates that energy based model training of the joint distribution improves calibration, robustness, and out-of-distribution detection while generating samples with better quality than GAN-based approaches. The reviewers are very excited about this work, and the energy-based perspective of generative and discriminative learning. There is a unanimous agreement to strongly accept this paper after author response.